# Estimating the Impact of Air Pollution on Healthcare-Seeking Behaviour by Applying a Difference-in-Differences Method to Syndromic Surveillance Data

**DOI:** 10.3390/ijerph19127097

**Published:** 2022-06-09

**Authors:** Roger Morbey, Gillian Smith, Karen Exley, André Charlett, Daniela de Angelis, Sally Harcourt, Felipe Gonzalez, Iain Lake, Alec Dobney, Alex Elliot

**Affiliations:** 1Real-Time Syndromic Surveillance Team, Field Service, UK Health Security Agency, Birmingham B2 4BH, UK; gillian.smith@ukhsa.gov.uk (G.S.); sally.harcourt@ukhsa.gov.uk (S.H.); alex.elliot@ukhsa.gov.uk (A.E.); 2Chemicals and Toxicology Science, UK Health Security Agency, London SE1 8UG, UK; karen.exley@ukhsa.gov.uk (K.E.); alec.dobney@ukhsa.gov.uk (A.D.); 3Statistics, Modelling and Economics, Analytics and Data Science, UK Health Security Agency, London NW9 5EQ, UK; andre.charlett@ukhsa.gov.uk (A.C.); daniela.deangelis@ukhsa.gov.uk (D.d.A.); 4School of Environmental Sciences, University of East Anglia, Norwich NR4 7TJ, UK; felipe.gonzalez@uea.ac.uk (F.G.); i.lake@uea.ac.uk (I.L.)

**Keywords:** public health, epidemiology, health burden, air pollution, syndromic surveillance

## Abstract

Syndromic surveillance data were used to estimate the direct impact of air pollution on healthcare-seeking behaviour, between 1 April 2012 and 31 December 2017. A difference-in-differences approach was used to control for spatial and temporal variations that were not due to air pollution and a meta-analysis was conducted to combine estimates from different pollution periods. Significant increases were found in general practitioner (GP) out-of-hours consultations, including a 98% increase (2–386, 95% confidence interval) in acute bronchitis and a 16% (3–30) increase in National Health Service (NHS) 111 calls for eye problems. However, the numbers involved are small; for instance, roughly one extra acute bronchitis consultation in a local authority on a day when air quality is poor. These results provide additional information for healthcare planners on the impacts of localised poor air quality. However, further work is required to identify the separate impact of different pollutants.

## 1. Introduction

Air pollution is a major cause of health problems across the world [1,2]. In addition to increased mortality, air pollution can increase morbidity with a range of symptoms, including: cough and wheezing [3], childhood asthma [4,5], and cardiopulmonary [6] and ischaemic heart disease [7]. Indeed, one study in Washington DC found that increases in ozone (O_3_) were associated with increased healthcare utilisation but not hospitalisations, whilst another study identified impacts on educational attainment, suggesting that monitoring just mortality or hospital admissions will underestimate the burden of impact [8,9]. Therefore, periods of poor air quality (when pollutant levels are increased) will lead to an increased demand for healthcare services [10,11,12].

Many countries routinely monitor air quality for the presence of pollutants and population health so that they can manage healthcare demand and take appropriate action, for example, communicating with at-risk groups [13]. The impact of infectious disease on population health is monitored through traditional laboratory surveillance and syndromic surveillance systems. Syndromic surveillance captures changes in symptom trends through monitoring electronic health records or other datasets [14]. Whilst laboratory surveillance is essential for identifying pathogens responsible for seasonal outbreaks of disease at the national, regional, and local level, syndromic surveillance can also monitor impacts on public health caused by environmental factors, such as air pollution [15,16,17]. However, it is unclear whether temporary increases in healthcare demand caused by localised air pollution can be detected by existing syndromic systems.

In this study, daily air pollutant and syndromic surveillance data were collected in England, United Kingdom (UK), to estimate the impact of periods of local poor air quality on healthcare demand. Furthermore, confounding factors that would bias estimates, including public holidays or deprivation, were accounted for. By using routine monitoring data, this research has direct relevance to surveillance systems, identifying which syndromic indicators are the most useful for the routine surveillance of air pollution.

## 2. Method

Data on air pollution were obtained from UK Air (uk-air.defra.gov.uk, accessed on 11 May 2019) for the following air pollutants: sulphur dioxide (SO_2_), nitrogen dioxide (NO_2_), ozone (O_3_), and particulate matter at 2.5 and 10 microns (PM_2.5_ and PM_10_). Daily data were collected between 1 April 2012 and 31 December 2017, from 96 English monitoring sites, covering 67 upper-tier local authorities (UTLAs) (England has 152 UTLAs, ranging in size from less than 100,000 to over 1 million population). These five pollutants are regularly monitored with daily forecasts to provide actions and health advice when levels are high enough to constitute a threat [18]. There are established daily air quality index (DAQI) thresholds in the UK describing low, moderate, high, or very high levels of pollutants [19]. For this study a location was defined as ‘exposed’ to an air pollutant if a monitoring site within the upper-tier local authority (UTLA) had recorded daily levels above the moderate threshold. By contrast a UTLA was labelled as a control if all sites in the authority had recorded daily levels below the moderate threshold with no missing data for any pollutant. The moderate DAQI threshold (break points) was taken as the boundary for exposure as this complies with existing recommendations for modifying behaviour for at-risk groups to protect from the health threat of air pollution. Therefore, these results will have direct relevance to current public health forecasting practice. A period of exposure was defined as having at least one day when pollutants were at a moderate or higher level and at least seven days prior to exposure when levels were consistently below the moderate threshold.

Syndromic data were obtained from three national Public Health England syndromic surveillance systems, including family doctor, known as general practitioners (GPs) in the UK, consultations (in hours and out of hours with unscheduled care), and calls to a National Health Service (NHS) telephone helpline (NHS 111) [20] Harcourt, 2012 #11 Harcourt, 2016 #846. Existing syndromic indicators which were routinely monitored in each system were selected based on expert clinical and epidemiological knowledge of those indicators which may be sensitive to air pollution, including: GP consultations for stroke, chest pain, eye irritation, cardiac problems, acute bronchitis, acute presenting asthma, difficulty breathing, pharyngitis, conjunctivitis, or allergic rhinitis; also, NHS 111 calls for sore throat, eye problems, difficulty breathing, and cough. In addition, as a sensitivity analysis one gastrointestinal indicator was included from each system (gastroenteritis from GP systems and 111 calls for diarrhoea), which should not be affected by air pollution.

To estimate the direct effects of air pollution on health indicators it was necessary to compare with controls where high levels of air pollution were not present. Controls can either be defined spatially, as other UTLAs during the same period that are not exposed, or as the same exposed UTLAs at different time periods (when not exposed). However, syndromic indicators may vary systematically between different locations and rates can change over time due to reasons other than air pollution. Therefore, in this study, a ‘difference-in-differences’ (DiD) method was applied to estimate and account for separately spatial and temporal effects and, thus, enable a direct estimate of the impact of air pollution. For each ‘period of exposure’ a UTLA was defined as a control if all its monitoring sites had levels below the moderate threshold, with no missing data either during the exposure period or for the 7 days before. The advantage of the DiD method is in accounting for spatial variables (e.g., deprivation) that cause differences between UTLAs which should not vary in the short term, and temporal variables (e.g., day of the week) that are likely to have similar effects across all UTLAs. This is important as it simplifies the analysis, particularly where variables are difficult to measure.

Each of the ‘periods of exposure’ was treated as a separate study, with regression models used to estimate the direct impact of exposure to air pollution. Furthermore, explanatory variables included lags of up to three days to allow for delays in their impact on pollution or syndromic data. Lags were investigated using separate models to allow for the possibility of a delay between exposure to pollution and presenting with health problems. Regression was carried out in R using the MASS package [21], using a negative binomial of syndromic count data with total system activity as an offset. Equation (1) shows the general form for models:(1)$$y_{st}=\ln\left(N_{st}\right)+E_{st}+L_{s}+P_{t}+X_{st}+D_{t}$$

The suffix, *s* represents the spatial dimension (UTLA) and *t* the temporal dimension (date). *y* is the daily count within the UTLA. *N* is the total activity in the syndromic system. *E* represents exposure to air pollution and is 1 for exposed UTLAs during the period of exposure, 0 otherwise. *E* gives the direct estimate for impact of air pollution. *L* represents location, 1 for those UTLAs that experienced levels above the moderate thresholds, 0 otherwise. L provides the estimate for differences between UTLAs that do not vary over time. *P* is 1 during the exposure period and 0 for the 7 days prior. *P* represents any temporal differences that are not due to air pollution. *X* is a matrix of other explanatory variables, including lags of up to 3 days. *D* is a factor for each date, to allow for day of week and other temporal effects.

Individual periods of exposure could be as short as one day and could involve a small number of UTLAs being exposed. Low numbers of data points result in considerable uncertainty for estimates of the direct effects of air pollution. Therefore, meta-analysis was used to combine the estimates and reduce the overall uncertainty. For each syndromic indicator, the estimates for each exposure period were combined to give a joint estimate for the impact of air pollution that was more precise. Estimates for each exposure period were inversely weighted by its standard deviation, thus, more weight was given to the more precise estimates. Separate meta-estimates were created for different possible lags between exposure and seeking healthcare as whilst it was not known whether lags would be important, it was likely that the same delays would apply to each period. Meta-analysis was appropriate for this study because, although different exposure periods would involve different UTLAs as cases or controls, the methodology is the same in each case.

## 3. Results

To create a directed acyclic graph (DAG) for the analysis, expert opinion and published literature were elicited to list many different factors that could have an impact on either the level of air pollution in a UTLA or the incidence of healthcare-seeking behaviour. These factors included socio-economic variables (e.g., deprivation), environmental factors (e.g., weather and pollen), respiratory pathogens (e.g., influenza), and the potential impact on healthcare-seeking behaviour from media reporting of pollution forecasts. A DAG was created using these variables to describe the understanding of their causal relationship (Figure 1). Many variables were considered unlikely to change in the short term, between exposure periods and the preceding control period of 7 days, e.g., deprivation or seasonality [22]. Therefore, these variables could be accounted for by measuring the difference between case and control UTLAs, both before and during the exposure periods. Additionally, some variables were likely to vary between the exposure period and during the previous day’s control period, e.g., public holidays but have similar effects across all UTLAs. Therefore, these variables could be accounted for by the estimates for the difference between the exposure period and preceding days across all UTLAs. The main remaining confounding factors potentially impacting on both pollution levels and syndromic data (and differing across time and space) were weather and pollen [23,24]. Therefore, explanatory weather variables were included based on daily rainfall (mm), humidity (%), and temperature (°C). In addition, temperature extremes were considered an important factor and, therefore, ‘temperature squared’ was included to allow for a more complex relationship than a simple linear variable [22]. The potential confounding effects of pollen and spores were accounted for by being included in the PM_10_ pollutants variable.

Pollutant data were available across all 2101 days included in the study, including over 193,000 site readings (Table 1). However, no readings were available where SO_2_ exceeded the moderate threshold and NO_2_ reached the moderate threshold of 201 μg/m^3^ in only one reading (Table 1).

It was not possible to calculate separate estimates for the impact of each type of pollutant. Firstly, there were no examples of SO_2_ or NO_2_ pollution during the study period that were above the moderate threshold. Secondly, periods with high PM_10_ pollution were found to be coincidental with PM_2.5_. PM_10_ and PM_2.5_ have a strong correlation as they are fractions of the same pollutant (often the same source), so high PM_10_ readings will normally occur in parallel to high PM_2.5_ (and same for the reverse). Finally, because one pollutant could act as a competing exposure when trying to estimate the impact of another pollutant [22], a UTLA was only included as a control when estimating the impact of O_3_ if corresponding data, confirming that its readings for particulate pollutants were below the moderate threshold, were available. Unfortunately, eliminating the possibility of one pollutant being a confounder when measuring another meant excluding a very high proportion of the data. Figure 2 shows the daily readings for O_3_, with non-grey markers showing data points that could be used to estimate the specific impact of O_3_. Whilst many days are excluded because levels everywhere were below the moderate threshold, most readings above the threshold could not be included because particulate levels might also be high in the same UTLA on the same day. Therefore, air pollution exposure was defined as a day when the moderate DAQI was exceeded for any pollutant. Table A1 lists the exposure periods and the number of UTLAs available for cases and controls. Further, 78 separate exposure periods were identified with pollutant levels above moderate thresholds; however, 24 of these could not be used because of missing exposure data during the preceding 7-day control period.

Syndromic data were available for every day of the study period, including a variable of ‘total activity’ for each surveillance system, which was used as a proxy for system coverage (as this can vary daily). Daily syndromic counts can vary considerably as underlying incidence of syndromes varies, some indicators are more specific than others, and coverage can vary by UTLA. The smallest mean daily count was 0.2 for GP out-of-hours stroke consultations and the highest 19.4 per 1000 registered patients for GP in-hours gastroenteritis consultations (Table 2).

Estimates for the direct impact of air pollution for individual exposure periods were calculated and visualised using forest plots. There was a lot of uncertainty and the 95% confidence intervals were wide. Figure 3 shows a forest plot for the individual estimates of each exposure period on GP out-of-hours consultations for acute bronchitis (with a lag of 0 days). The meta-analysis estimate combining all periods has a much shorter confidence interval, representing the greater certainty achieved by combining data from all periods.

The meta-analysis resulted in estimates for the rate ratio (RR) of the direct effect of air pollution, which ranged from RR 0.55 (0.12–2.28) for same day GP out-of-hours (GPOOH) stroke consultations, to RR 1.98 (1.02–3.86) for same day GP out-of-hours acute bronchitis consultations. All confidence intervals in this study were calculated to be statistically significant at a 95% level. Table 2 shows the lags with the highest estimates for each indicator, with all lags available in Table A3. The highest RR was GPOOH acute bronchitis with the next highest estimate GP out-of-hours stroke consultations at a 1-day lag (RR 1.45 [CI 0.40–5.34]; however, the uncertainty around GPOOH stroke estimates was very high (Table 2). Statistically significant estimates were obtained for other GPOOH indicators, including cardiac consultations (0-day lag); acute presenting asthma, wheeze, and difficulty breathing problems (+2-day lag); chest pain (+3-day lag); and eye irritation (+4-day lag). There was also a significant association between air pollution and NHS 111 calls for eye problems at both 0- and +1-day lag. Estimates for the impact on GP in-hours consultations were smaller and none of them were statistically significant at the 95% level.

Comparisons for the meta-analysis across all syndromes are shown visually in a Forest plot, Figure 4. Whilst some of the highest effects were measured for GPOOH syndromes, the sparse data resulted in much wider confidence intervals, as depicted by smaller box sizes.

## 4. Discussion

In this paper the direct impact of air pollution on the number of patients presenting with a range of different symptoms was estimated. Many of the symptoms included in this study (e.g., difficulty breathing and eye irritation) can have multiple aetiologies, including infectious diseases and environmental factors other than air pollution, e.g., pollen and spores. It is, therefore, important to try and disentangle the direct impact of air pollution from other confounding variables to understand its contribution. Historically, researchers built predictive models, including confounding variables, to assess the relative contribution of air pollution. However, these predictive models are limited in their interpretation for individual coefficients. For example, the impact of an increase in air pollution can only be compared to the idealised, and potentially unrealistic, scenario where all other coefficients are kept constant. Therefore, there has been a lot of interest in the development of causal inference methods to provide estimates of the direct effects of air pollution [25]. The approach used in this study was to use two causal inference techniques, DAGs and DiD.

One of the main problems in estimating the impact of poor air quality using healthcare data is the number of other variables that may also trigger similar clinical presentations or symptoms, for instance, seasonal influenza epidemics. Furthermore, seasonal and socio-economic factors will result in temporal and spatial variations in syndromic data. Therefore, DAGs [25] were used to identify variables that could bias the estimates for the impact of air pollution. Using DAGs helps to formalise the assumptions behind a study and identify the role of different variables. For instance, do variables affect both the exposure (air pollution) and the outcome (syndromic data)? If so, they are ‘confounders’ or just affect the outcome as ‘competing exposures’. Importantly, failing to account for confounders can lead to bias in estimates for direct effects (of air pollution), whilst competing exposures can increase the uncertainty around the size of estimates. Furthermore, if there had been any variables that acted as ‘mediators’ between the exposure and outcome, these would have to be excluded from the analysis to avoid introducing ‘collider bias’ to the estimates of the direct effects of air pollution [26].

The DAG enabled consideration and identification of the causal effects of various factors, which were either stable in the short term (e.g., deprivation) or stable across location (e.g., public holidays in England). Consequently, by applying a DiD method to compare cases with both control locations and control time periods, it was possible to account for many of the confounding factors. Finally, the DAG identified the factors that were not stable across time or space (e.g., weather) and, therefore, needed to be included in regression models.

The object of public health surveillance is to provide useful information for action. Therefore, we used routinely available data, including meteorological metrics and ongoing syndromic surveillance systems. Consequently, the results can validate the usefulness of syndromic surveillance for monitoring the impact of days of poor air quality, as currently measured using the DAQI. Furthermore, the syndromic indicators give an estimate for the impact on health services of air pollution, enabling pollution forecasts to be turned into forecasts of changes in demand for health services.

The largest estimated increase in syndromic rates due to poor air quality was 1.98 (1.02, 3.86) for the GP out-of-hours acute bronchitis indicator. A typical UTLA with good coverage will record around 430 GPOOH consultations a day, of which 0.7 will have acute bronchitis and 8.2 cardiac symptoms. This central estimate for the impact of a day with poor air quality would translate to an extra 0.7 (0.0, 2.0 95% CI) acute bronchitis consultations and an extra 2.1 (0.5, 4.0 95% CI) cardiac patients. However, days with poor air quality were relatively rare in the period studied.

A range of syndromic indicators were compared across three syndromic systems. Almost all the significant increases were detected in the GP out-of-hours system; no GP in-hours syndromes giving increases were statistically significant. This could reflect differences in the case mix of patients or the availability of services. GP in-hours services require booking an appointment in advance, limiting the availability of same-day consultations, and are only open during office hours. By contrast, GP out-of-hours services are available at weekends and evenings. It is likely, that patients use these services differently, perhaps using out-of-hours for urgent acute health needs and using in-hours for chronic disease management. Similar syndromic indicators in different systems have similar names but different underlying codes; for example, the ‘asthma/wheeze/difficulty breathing’ GPOOH indictor represents different conditions to the GPIHSS ‘acute presenting asthma’ indicator. It is unlikely, that the lack of significant results for GPIHSS indicators is due to data volume, as coverage is greater than for GPOOH. Significant increases were found for a range of GPOOH indicators, including respiratory, cardiac, and eye irritation syndromes. However, for NHS 111 calls, the only significant indicator was for eye problems.

The importance of the impact of air pollution on public health is reflected in the number and range of studies in recent years. Bae et al. reviewed research papers in South Korea into the mortality and morbidity of air pollution [27]; symptoms of morbidity included asthma, respiratory and cardiovascular hospitalization, upper and lower respiratory symptoms, low birth weight, depression, insulin resistance, allergic diseases, airway hyperresponsiveness, and new episodes of wheezing. A similar country-wide review found evidence of increased respiratory and cardiovascular disease in China associated with air pollution [1].

An important limitation in population studies, particularly involving syndromic data, is that no direct link can be shown between exposure and morbidity. There is uncertainty around whether the patients presenting with symptoms have been exposed to high levels of the pollutants recorded in their locations and, therefore, a causal link for individuals cannot be proven. Similarly, there may be very complex causal interactions, for example, between weather and pollutant variables, that have not been accounted for. Finally, there is always the possibility of other unknown or unmeasured variables, e.g., indoor pollutants that could bias results.

Whilst many of the studies mentioned in the literature reviews above provide estimates for specific pollutants, this study has not attempted to distinguish between the impacts of different pollutants. Primarily, this was due to limitations in the available data, with no examples of periods with high levels of nitrogen dioxide NO_2_ or sulphur dioxide SO_2_. It was not possible to disentangle the impact of PM_10_ from PM_2.5_ because high levels of these pollutants usually occurred concurrently in both time and place. Finally, a conservative approach to missing data was taken, only including UTLAs as controls when there was evidence of low pollutant levels for particulate matter and O_3_. Thus, there were limited examples available and insufficient data to provide separate estimates for O_3_ and particulate matter.

To validate the approach, a sensitivity analysis included gastrointestinal indicators, which would not normally be affected by air pollution. GPOOH gastroenteritis consultations had an estimate that was just statistically significant at the 95% level. This was the only gastrointestinal indicator with a significant result. It suggests that results should be interpreted with some caution, particularly estimates for indicators where the increases are small and the confidence intervals are wide. However, it does not invalidate the estimates for GPOOH acute bronchitis or cardiac indicators, which were considerably larger than gastroenteritis.

The methodology was chosen to address the limitations in the study. The causal inference DiD method addressed the issues of confounders and bias due to the many factors affecting morbidity. Furthermore, this approach makes modelling assumptions explicit and transparent and provides a clearer interpretation of results than is possible with predictive modelling approaches. Similarly, the use of meta-analysis enabled more precise estimates to be obtained by combining results across different periods of poor air quality in different locations. Meta-analysis has often been criticised for combining results that are not comparable, but it is appropriate in this setting, where the same methodology and data collection is used throughout.

The impacts of air pollution across three national syndromic surveillance systems in England were studied. In future, this analysis could be repeated across the newer English syndromic systems, incorporating data from emergency departments, ambulance dispatch calls, and an online symptom checker. These systems would provide additional information on a different case-mix of patients, including those with more severe symptoms who need to attend hospital. Furthermore, the difference-in-differences approach used here to compare local areas with each other and with control periods could be extended into a more general surveillance tool to measure the impact of exposures with a wide range of causes.

## 5. Conclusions

The research has shown that syndromic systems are sensitive to the local impacts of air pollution, with increases seen in a range of indicators, including respiratory, cardiac, and eye irritation. Furthermore, the best indicators are likely to be from the GP out-of-hours (not GP in-hours) system. This information will have a direct impact on how syndromic surveillance is conducted and the interpretation of real-time surveillance for decision makers during air pollution episodes. Further, for some indicators, lags can be expected for up to four days, which again, are useful for interpreting daily statistical alarms following forecasts of periods of poor air quality. Finally, there is increasing interest in local authorities producing health warnings to their residents when air pollution is forecast, and it would be possible to use this approach to evaluate and monitor the impact of these warnings.

## Figures and Tables

**Figure 1 ijerph-19-07097-f001:**
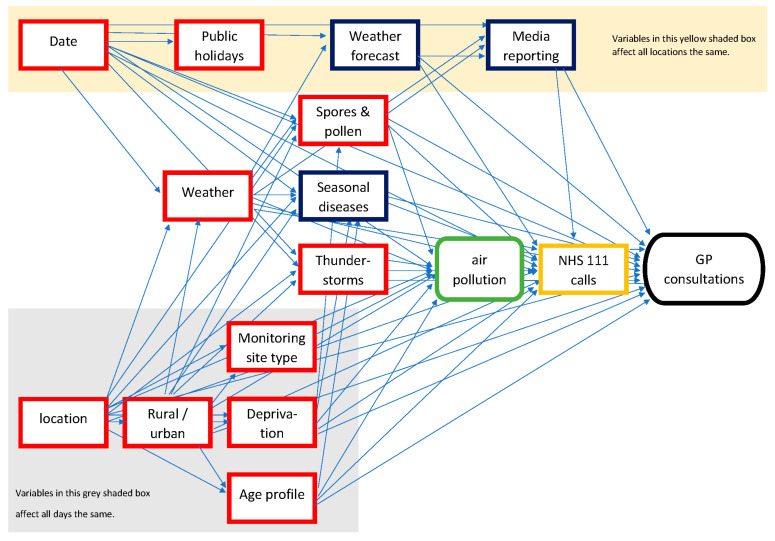
DAG to investigate impact of air pollution on GP consultations. Red boxes depict confounding variables that affect both the exposure of air pollution and the outcome, GP consultations. Blue boxes show variables that are competing exposures, affecting GP consultations but not pollution.

**Figure 2 ijerph-19-07097-f002:**
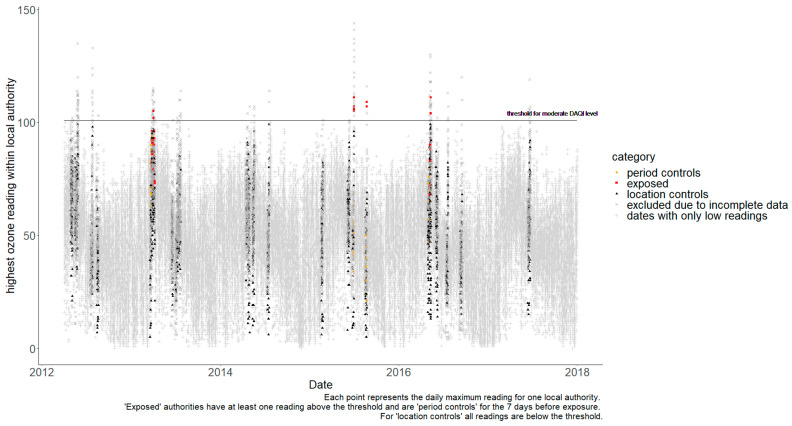
Daily maximum readings for O_3_ by local authority.

**Figure 3 ijerph-19-07097-f003:**
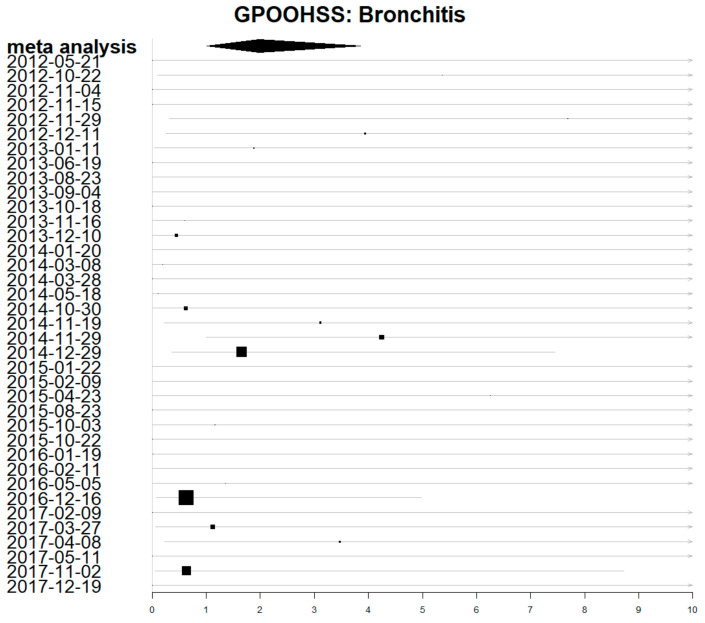
Forest plot showing estimates of rate ratio for the impact of air pollution on GP out-of-hours acute bronchitis consultations. Dates are the first day of each exposure period. Larger boxes depict more precise estimates which contribute more weight.

**Figure 4 ijerph-19-07097-f004:**
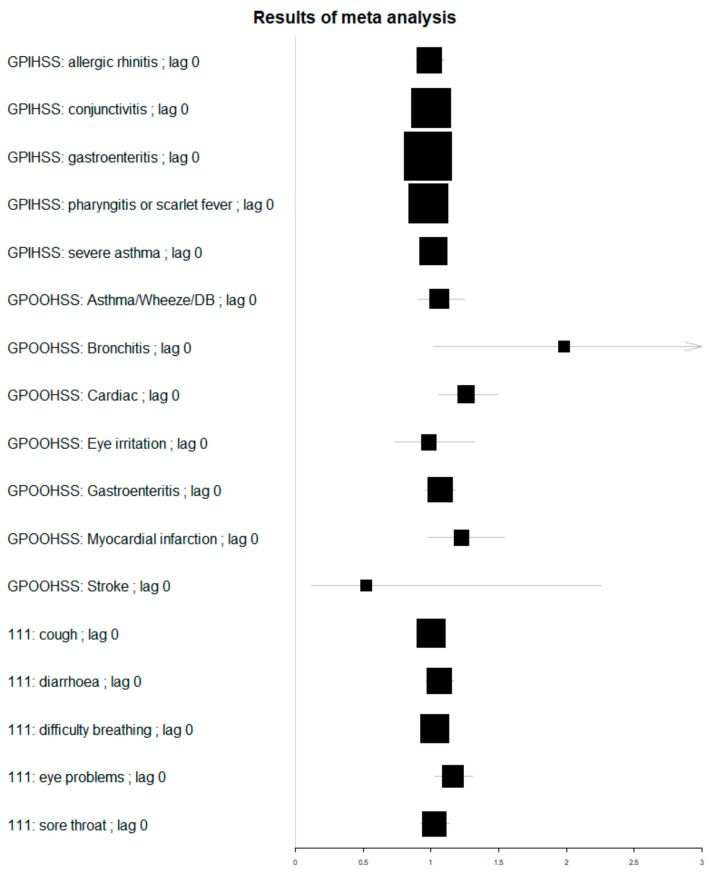
Forest plot of estimates for rate ratio of impact of air pollution on syndromic indicators with a zero lag. Box size is proportional to estimate precision.

**Table 1 ijerph-19-07097-t001:** Summary of pollutant data illustrating the threshold levels and the number of days when these were exceeded during the study period (1 April 2012–31 December 2017).

	Pollutant	Daily Air Quality Index (DAQI)			Max Value Recorded
		Moderate	High	Very High	
Lower threshold for DAQI level (μg/m^3^)	NO_2_	201	401	601	201
	O_3_	101	161	241	144
	PM_10_	51	76	101	210
	PM_2.5_	36	54	71	102
	SO_2_	267	533	1065	73
		Moderate	High	Very High	All levels
Number of site readings within DAQI band	NO_2_	1	0	0	193,098
	O_3_	196	0	0	114,952
	PM_10_	1109	98	7	83,888
	PM_2.5_	2006	345	57	99,725
	SO_2_	0	0	0	36,657
Number of unique days with a reading within DAQI band	NO_2_	1	0	0	2101
	O_3_	67	0	0	2101
	PM_10_	215	32	6	2101
	PM_2.5_	212	51	17	2101
	SO_2_	0	0	0	2101

Pollutants: sulphur dioxide (SO_2_), nitrogen dioxide (NO_2_), ozone (O_3_), and particulate matter at 2.5 and 10 microns (PM_2.5_ and PM_10_).

**Table 2 ijerph-19-07097-t002:** Meta analysis results—estimate of rate ratios for direct effect of air pollution on syndromic indicators. Results where lower 95% confidence interval is at least 1 are highlighted in bold.

System: Syndrome	Lag (Days)	Rate Ratio	95% Confidence Interval	
**GPOOH: acute bronchitis**	**0**	**1.98**	**1.02**	**3.86**
GPOOH: Stroke	1	1.45	0.40	5.34
**GPOOH: Eye irritation**	**4**	**1.35**	**1.01**	**1.80**
**GPOOH: chest pain**	**3**	**1.27**	**1.01**	**1.58**
**GPOOH: Cardiac**	**0**	**1.26**	**1.06**	**1.49**
**GPOOH: Asthma/Wheeze/DB**	**2**	**1.17**	**1.00**	**1.35**
**NHS 111: eye problems**	**0**	**1.16**	**1.03**	**1.30**
**GPOOH: Gastroenteritis**	**1**	**1.13**	**1.02**	**1.25**
GPIHSS: allergic rhinitis	3	1.07	0.96	1.20
NHS 111: sore throat	1	1.07	0.97	1.19
GPIHSS: acute presenting asthma	2	1.06	0.95	1.17
NHS 111: diarrhoea	4	1.06	0.96	1.16
GPIHSS: pharyngitis or scarlet fever	3	1.04	0.98	1.10
NHS 111: cough	1	1.04	0.95	1.13
NHS 111: difficulty breathing	0	1.03	0.95	1.12
GPIHSS: gastroenteritis	4	1.02	0.98	1.07
GPIHSS: conjunctivitis	0	1.00	0.95	1.06

Syndromic Surveillance Systems: GP out-of-hours and unscheduled care (GPOOH), GP in hours (GPIHSS), National Health Service telephone helpline (NHS 111).

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
