# Peer review of "Estimating the Impact of Air Pollution on Healthcare-Seeking Behaviour by Applying a Difference-in-Differences Method to Syndromic Surveillance Data"

_ijerph, 2022, doi:10.3390/ijerph19127097_

Round 1

Reviewer 1 Report

In this manuscript, several concerns need to be addressed as follows:

  1. Line 7: the first affiliation is unclear. Please, give full details.
  2. The manuscript needs to be revised for the English and the overall style of writing. The writing style should be formal from the third-person perspective. Do not use we or our. Also, the sentence should not begin with abbreviations like that in line 57 " DAGs" and lines 46 in the abstract AF.
  3. There is a problem in using abbreviations throughout the manuscript. The full term should be mentioned first with the abbreviation between paresis then the abbreviations should be used throughout the manuscript. E.g., in line 16, GP should be replaced by the general practitioner (GP) then the abbreviation should be used further. Such errors have been repeated for most abbreviations throughout the manuscript.
  4. More keywords should be added like air pollution…etc.
  5. Line 52-62: should be transferred to the introduction or discussion section.
  6. Figure 1: the text should be enlarged within the figure as it is unclear.
  7. Tables 1 and 2: the full term of all abbreviations used within the table should be clarified in the footnote.
  8. The details of Figures 2 and 4 are unclear. Please, reconstruct.

Author Response

  1. Affiliations have been updated to include directorate within UK Health Security Agency for authors who work for UKHSA.
  2. The document has been changed so that the third person tense is used throughout. Sentences have been changed so that they do not start with an abbreviation.
  3. Abbreviations are now explained at their first occurrence throughout.
  4. Air pollution and syndromic surveillance have been added to the key words
  5. The paragraph discussing directed acyclic graphs has been moved from methods to discussion.
  6. We have re-created figure 1 to make the text clearer. It is also in a more generalised form so it does not just apply to asthma consultations.
  7. Footnotes have been added to tables 1 & 2 to explain abbreviations
  8. The text in figures has been increased to improve legibility.

Reviewer 2 Report

The reviewer appreciated the authors comments on limitations of the study.  The study seemed to be designed carefully.  The task is difficult and the thought process could be applied by other researchers to expand on the findings of this paper. 

Only comments- Figure 2 is difficult to read.  The x-axis on Figure 3 can have the font increased. Overall- good job.  Maybe other researchers may have data that would allow the separation of estimates each pollutant.

Author Response

The text in figures has been increased to improve legibility.

Reviewer 3 Report

What are the strengths of the study: Study of larger sample size than other manuscripts studying this question. What were your major concerns: The methodology is still difficult to interpret. The results do not seems to support the conclusions by the authors.

 Additional Comments to Authors

  1. please clear the rationale and detailed process to get directed acyclic graphs (DAG), and clarify the definition of  ‘competing exposures’ and ‘confounders’, Why only PM10 involved in DAG analysis?
  2. line118, ‘y’ is the daily count for syndromic data in a UTLA, so what daily count for what?
  3. line67, Write the full name of UTLAs as it is first mentioned
  4. line228,why there lag days (1-4day)?not 5 or more days?
  5. Figure4 showed Forest plot of estimates for rate ratio of impact of air pollution on syndromic indicators on the same day, which day? please add the Interpretation of results of Figure4 in ‘result’ section.
  6. make clear the each variable in Equation 1, it is highly de-normalized

Author Response

  1. We have added more detail about the construction of the DAG in start of the results section. More detail about variables types has been added, along with a reference in the discussion. Figure 1 has been modified to a more generalised form so it is clear that the approach does not just apply to PM10s.
  2. The equation is a generalised form used for any syndrome, the text has been changed to make this less confusing.
  3. Abbreviations are now explained at their first occurrence throughout.
  4. The authors considered that acute effects of air pollution were unlikely to occur more than four days after a period of high air pollution.
  5. The caption for figure 4 has been clarified so that it is clear that these are the results for zero-lag models. Interpretation of the figure 4 results has been added to the results.
  6. The authors agree that the variables in equation 1 are not normalised, regression was done using a negative binomial model to account for count data that may be subject to over-dispersion.

The text in figures has been increased to improve legibility.

Round 2

Reviewer 1 Report

No further comments to be addressed

Author Response

Thank you for your help with reviewing this paper.

Reviewer 3 Report

none

Author Response

(The authors gave the same response as above.)
